# On The Classification-Distortion-Perception Tradeoff

**Dong Liu, Haochen Zhang, Zhiwei Xiong**
University of Science and Technology of China, Hefei 230027, China
dongeliu@ustc.edu.cn

## Abstract

Signal degradation is ubiquitous, and computational restoration of degraded signal has been investigated for many years. Recently, it is reported that the capability of signal restoration is fundamentally limited by the so-called perception-distortion tradeoff, i.e. the distortion and the perceptual difference between the restored signal and the ideal "original" signal cannot be made both minimal simultaneously. Distortion corresponds to signal fidelity and perceptual difference corresponds to perceptual naturalness, both of which are important metrics in practice. Besides, there is another dimension worthy of consideration–the semantic quality of the restored signal, i.e. the utility of the signal for recognition purpose. In this paper, we extend the previous perception-distortion tradeoff to the case of classification-distortion-perception (CDP) tradeoff, where we introduced the classification error rate of the restored signal in addition to distortion and perceptual difference. In particular, we consider the classification error rate achieved on the restored signal using a predefined classifier as a representative metric for semantic quality. We rigorously prove the existence of the CDP tradeoff, i.e. the distortion, perceptual difference, and classification error rate cannot be made all minimal simultaneously. We also provide both simulation and experimental results to showcase the CDP tradeoff. Our findings can be useful especially for computer vision research where some low-level vision tasks (signal restoration) serve for high-level vision tasks (visual understanding). Our code and models have been published.

## 1 Introduction

Signal degradation refers to the corruption of the signal due to many different reasons such as interference and the blend of interested signal and uninterested signal or noise, which is observed ubiquitously in practical information systems. The cause of signal degradation may be physical factors, such as the imperfectness of data acquisition devices and the noise in data transmission medium; or may be artificial factors, such as the lossy data compression and the transmission of multiple sources over the same medium at the same time. In addition, in cases where we want to enhance signal, we may assume the signal to have been somehow "degraded." For example, if we want to enhance the resolution of an image, we assume the image is a degraded version of an ideal "original" image that has high resolution [6].

To tackle signal degradation or to fulfill signal enhancement, computational restoration of degraded signal has been investigated for many years. There are various signal restoration tasks corresponding to different degradation reasons. Taken image as example, image denoising [23], image deblurring [17], single image super-resolution [6], image contrast enhancement [7], image compression artifact removal [5], image inpainting [22], . . . , all belong to image restoration tasks.

Different restoration tasks have various objectives. Some tasks may be keen to recover the "original" signal as faithfully as possible, like image denoising is to recover the noise-free image, compression artifact removal is to recover the uncompressed image. Some other tasks may concern more about the perceptual quality of the restored signal, like image super-resolution is to produce image details

to make the enhanced image look like having "high-resolution," image inpainting is to generate a complete image that looks "natural." Yet some other tasks may serve for recognition or understanding purpose: for one example, an image containing a car license plate may have blur, and image deblurring can achieve a less blurred image so as to recognize the license plate [13]; for another example, an image taken at night is difficult to identify, and image contrast enhancement can produce a more naturally looking image that is better understood [10]. Recent years have witnessed more and more efforts about the last category [16, 19].

Given the different objectives, it is apparent that a signal restoration method designed for one specific task shall be evaluated with the specific metric that corresponds to the task's objective. Indeed, the aforementioned objectives correspond to three groups of evaluation metrics:

1. *Signal fidelity metrics* that evaluate how similar is the restored signal to the "original" signal. These include all the full-reference quality metrics, such as the well-known mean-squared-error (MSE) and its counterpart peak signal-to-noise ratio (PSNR), the structural similarity (SSIM) [21], and the difference in features extracted from original signal and restored signal [8], to name a few.

2. *Perceptual naturalness metrics* that evaluate how "natural" is the restored signal with respect to human perception. These are usually known as no-reference quality metrics [14, 15]. Recently, the popularity of generative adversarial network (GAN) has motivated a mathematical formulation of perceptual naturalness [3].

3. *Semantic quality metrics* that evaluate how "useful" is the restored signal in the sense that it better serves for the following semantic-related analyses. For example, whether a restored sample can be correctly classified is a measure of the semantic quality. There are only a few studies about semantic quality assessment methods [12].

It is worth noting that signal fidelity metrics have dominated in the research of signal restoration. However, is one method optimized for signal fidelity also optimal for perceptual naturalness or semantic quality? This question has been overlooked for a long while until recently. Blau and Michaeli considered signal fidelity and perceptual naturalness, and concluded that optimizing for the two metrics can be *contradictory* [3]. Indeed, they provided a rigorous proof of the existence of the perception-distortion tradeoff: with distortion representing signal fidelity and perceptual difference representing perceptual naturalness, one signal restoration method cannot achieve both low distortion and low perceptual difference *up to a bound*. This conclusion reveals the fundamental limit of the capability of signal restoration, and quickly inspires the investigation of perceptual naturalness metrics in different tasks [2, 20].

Following the work of the perception-distortion tradeoff, in this paper, we aim to consider the three groups of metrics jointly, i.e. we want to study the relation between signal fidelity, perceptual naturalness, and semantic quality, in the context of signal restoration. In particular, we consider *classification error rate* as a representative of semantic quality metrics, because classification is the most fundamental semantic-related analysis. We adopt the classification error rate achieved on the restored signal using a predefined classifier as the third dimension in addition to distortion and perceptual difference. We provide a rigorous proof of the existence of the classification-distortion-perception (**CDP**) tradeoff, i.e. the distortion, perceptual difference, and classification error rate cannot be made all minimal simultaneously. We also provide both simulation and experimental results to showcase the CDP tradeoff. Our code and models are published at `https://github.com/AlanZhang1995/CDP-Tradeoff`.

To the best of our knowledge, this paper is the first to reveal the fundamental tradeoff between the three kinds of quality metrics: signal fidelity, perceptual naturalness, and semantic quality, in the context of signal restoration. Our results imply that, if a signal restoration method is meant to serve for recognition or understanding purpose, then the method is better optimized for semantic quality instead of signal fidelity or perceptual naturalness. This is *in contrast to* most of the existing practices. It then calls for more investigation of semantic quality metrics.

## 2 Problem Formulation

Consider the process: $X \rightarrow Y \rightarrow \hat{X}$, where $X$ denotes the ideal "original" signal, $Y$ denotes the degraded signal, and $\hat{X}$ denotes the restored signal. We consider $X$, $Y$, and $\hat{X}$ each as a discrete

random variable. The cases of continuous random variables can be deduced in a similar manner, and are omitted hereafter. The probability mass function of $X$ is denoted by $p_X(x)$. The degradation model is denoted by $P_{Y|X}$, which is characterized by a conditional mass function $p(y|x)$. The restoration method is then denoted by $P_{\hat{X}|Y}$ and characterized by $p(\hat{x}|y)$.

## 2.1 Distortion, Perceptual Difference, and Classification Error Rate

There are different categories of quality metrics to evaluate the signal restoration methods. For the first category, signal fidelity, we usually adopt distortion that is defined precisely as the expectation of a given bivariate function, i.e.

$$\text{Distortion} := \mathbb{E}[\Delta(X, \hat{X})] \tag{1}$$

where $\mathbb{E}$ is to take expectation over the joint distribution $p_{X,\hat{X}}$, $\Delta(\cdot, \cdot) : \mathcal{X} \times \hat{\mathcal{X}} \to \mathbb{R}^+$ is a given function to measure the difference between the original and the restored samples. This definition is corresponding to the common practice of using various forms of full-reference loss functions, e.g. MSE, in the signal restoration tasks. The definition measures the dissimilarity, i.e. the lower the better, while some popular quality metrics such as PSNR and SSIM measure similarity.

For the second category, perceptual naturalness, it has been proved in [3] that the perceptual quality evaluated by human when performing a real-or-fake test is indeed equivalent to the total-variation (TV) distance between the distribution of the original signal and that of the restored signal. Following [3], we define perceptual difference as

$$\text{Perceptual Difference} := d(p_X, p_{\hat{X}}) \tag{2}$$

where $d(\cdot, \cdot)$ is a function to measure the difference between two probability mass functions, such as the TV distance and the Kullback-Leibler (KL) distance. Perception is also the lower the better.

For the third category, semantic quality, we will focus on the classification error rate achieved on the restored signal using a predefined classifier in this paper. We will discuss the case of classifying the signal into two categories, and note that extension to multiple categories is straightforward.

We assume each sample of the original signal belongs to one of two classes: $\omega_1$ or $\omega_2$. The *a priori* probabilities and the conditional mass functions are assumed to be known as $P_1, P_2 = 1 - P_1$ and $p_{X1}(x), p_{X2}(x)$, respectively. In other words, $X$ follows a two-component mixture model: $p_X(x) = P_1 p_{X1}(x) + P_2 p_{X2}(x)$. Accordingly, $Y$ follows the model: $p_Y(y) = P_1 p_{Y1}(y) + P_2 p_{Y2}(y)$, and $\hat{X}$ follows the model: $p_{\hat{X}}(\hat{x}) = P_1 p_{\hat{X}1}(\hat{x}) + P_2 p_{\hat{X}2}(\hat{x})$, where

$$p_{Yi}(y) = \sum_{x \in \mathcal{X}} p(y|x) p_{Xi}(x), i = 1, 2 \tag{3}$$

$$p_{\hat{X}i}(\hat{x}) = \sum_{y \in \mathcal{Y}} p(\hat{x}|y) p_{Yi}(y) = \sum_y \sum_x p(\hat{x}|y) p(y|x) p_{Xi}(x), i = 1, 2 \tag{4}$$

A binary classifier can be denoted by

$$c(t) = c(t|\mathcal{R}) = \begin{cases} \omega_1, & \text{if } t \in \mathcal{R} \\ \omega_2, & \text{otherwise} \end{cases} \tag{5}$$

If we apply this classifier on the restored signal $\hat{X}$, we shall achieve an error rate

$$\text{Classification Error Rate} := \varepsilon(\hat{X}|c) = \varepsilon(\hat{X}|\mathcal{R}) = P_2 \sum_{\hat{x} \in \mathcal{R}} p_{\hat{X}2}(\hat{x}) + P_1 \sum_{\hat{x} \notin \mathcal{R}} p_{\hat{X}1}(\hat{x}) \tag{6}$$

## 2.2 The CDP Function

We are now ready to define the CDP function, which is the focus of our study.

**Definition 1.** *The classification-distortion-perception (CDP) function is*

$$C(D, P) = \min_{P_{\hat{X}|Y}} \varepsilon(\hat{X}|c_0), \textit{subject to } \mathbb{E}[\Delta(X, \hat{X})] \le D, d(p_X, p_{\hat{X}}) \le P \tag{7}$$

*where $c_0 = c(\cdot|\mathcal{R}_0)$ is a predefined binary classifier.*

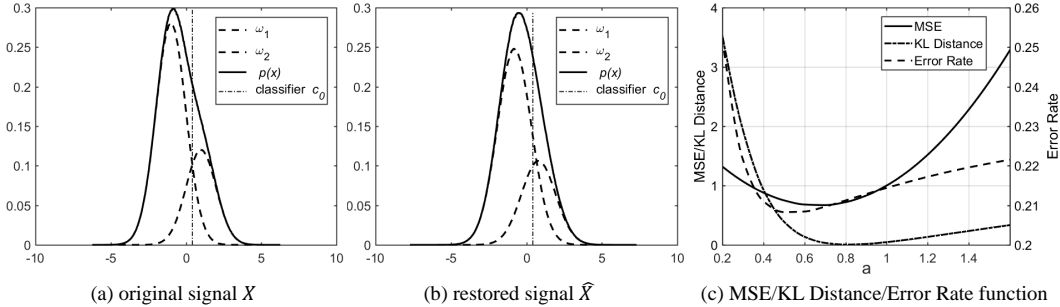

|       (a) original signal $X$       |       (b) restored signal $\widehat{X}$       |       (c) MSE/KL Distance/Error Rate function       |

Figure 1: A toy example to showcase the CDP function. See text for details.

The CDP function characterizes how well a signal restoration method $(P_{\hat{X}|Y})$ can perform, if it is constrained to have a low distortion (less than $D$) and a low perceptual difference (less than $P$). Note that if $D = \infty, P = \infty$, then the restoration method is optimized purely for lowering error rate, and thus the CDP function reaches its minimum. By defining the CDP function, we are interested to know whether a constrained optimization can perform as well as an unconstrained one. This is because the optimization for distortion has been studied extensively, and if the optimization for distortion or perception also leads to the optimization for classification, then we are done. However, this is not the case, as we will prove.

Another issue is about the *predefined* classifier in the definition of the CDP function. One may be curious to know whether it is possible to adjust the classifier itself to achieve a lower error rate: this is surely possible. However, there are a practical difficulty to train the optimal classifier for the restored signal, since the distribution of the restored signal is dependent on the restoration method that is to be decided. Next, we may ask whether it is practical to adjust the restoration method and the classifier simultaneously. However, this is not necessary, because we can prove that the optimal classifier for $\hat{X}$ cannot outperform the optimal classifier for $Y$ (*see the supplementary for the proof*). That says, we do not need to perform signal restoration if we can train the optimal classifier for the degraded signal. But this is another practical difficulty: the distribution of the signal to be restored is often unknown (called blind restoration), so it is not easy to train the optimal classifier for it. In summary, if dealing with blind restoration, i.e. the distribution of the degraded signal is unknown, then it is difficult to achieve the optimal classifier for either degraded or restored signal, so using a predefined classifier is a more practical choice. If dealing with non-blind restoration, i.e. the distribution of the degraded signal is known, then we can achieve the optimal classifier for the degraded signal, and it is not necessary to perform signal restoration prior to classification as it will not improve the classification performance. In this paper, we consider the case of blind restoration, and we leave the case of non-blind restoration as our future work.

### 2.3 Toy Example

To showcase the characteristic of the CDP function, we conduct simulations with a toy example. As shown in Figure 1, the original signal follows a two-component Gaussian mixture model: $P_1 = 0.7, P_2 = 0.3, p_{X1}(x) = \mathcal{N}(-1, 1), p_{X2}(x) = \mathcal{N}(1, 1)$. The signal is corrupted by additive white Gaussian noise: $Y = X + N$, where $N \sim \mathcal{N}(0, 1)$. The denoising method is linear: $\hat{X} = aY$ where $a$ is an adjustable parameter. For example, the restored signal with $a = 0.8$ is depicted in Figure 1 (b). We use the binary classifier that is the optimal for the original signal to evaluate error rate. In addition, we use MSE to evaluate distortion, and use the KL distance to evaluate perception. Under these settings, we can derive closed-form functions of MSE and error rate with respect to the parameter $a$ (*see the supplementary for details*). For the KL distance, we do not have closed-form expression so we perform numerical calculation. We then use numerical methods to calculate the CDP function and depict the function in Figure 2.

First, the CDP function is monotonically non-increasing, i.e. the minimal attainable error rate decreases as the maximal allowable distortion and perception increase. It implies that if one wants to have a restoration method for better classification performance, it must come at the cost of higher distortion, lower perceptual quality, or both. Second, the CDP function is convex, indicating that

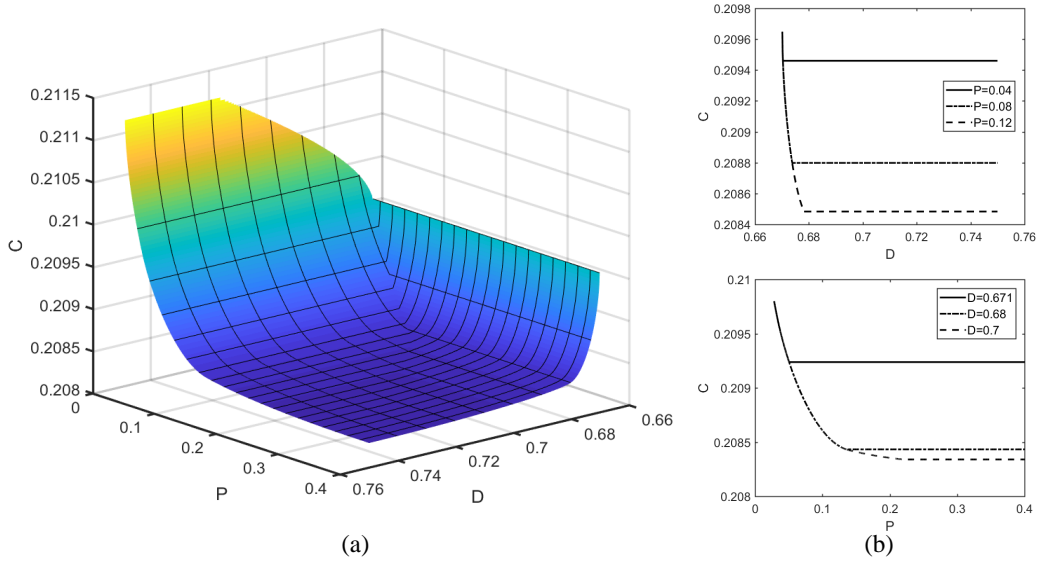

<div align="center">(a)                  (b)</div>

Figure 2: (a) The CDP function for the toy example where we can find the minimal attainable error rate (C) decreases as the maximal allowable MSE (D) and KL divergence (P) increase. (b) Profiles of the CDP function at different $P$ values and $D$ values respectively, from which we can find the function is convex.

if $D$ ($P$) is smaller, the error rate increases faster. Thus, minimizing $D$ ($P$) can be quite harmful for the classification performance. Third, from Figure 2 (b), we observe that when $D$ is small, $C$ is invariant with $P$, and when $D$ is large, $C$ is invariant with $D$. In this example, the feasible domain of $P_{\hat{X}|Y}$ is fully determined by the feasible set of $a$, which is indeed the intersection of the feasible set of $a$ defined by $D$ and that defined by $P$. If $D$ is small, the feasible set defined by $D$ is also small and determines the intersection. If $D$ is large, the feasible set is also large and has no effect on the intersection. Similarly, from Figure 2 (b), we observe that when $P$ is small, $C$ is invariant with $D$, and when $P$ is large, $C$ is invariant with $P$. It can be interpreted similarly. Last but not the least, note that the areas where $D$ and $P$ are both small are not present in the CDP function, which results from the perception-distortion tradeoff [3].

In more general situations, it is impossible to solve Eq. (7) analytically. But some properties of the CDP function are still valid, as shown in the following section.

## 3 The CDP Tradeoff

**Theorem 1.** *Considering the CDP function (7), if $d(\cdot, q)$ is convex in $q$, then $C(D, P)$ is*

1. *monotonically non-increasing,*

2. *convex in $D$ and $P$.*

*Proof.* For the first point, simply note that when increasing $D$ or $P$, the feasible domain of $P_{\hat{X}|Y}$ is enlarged; as $C(D, P)$ is the minimal value of $\varepsilon(\hat{X}|c_0)$ over the feasible domain, and the feasible domain is enlarged, the minimal value will not increase.

For the second point, it is equivalent to prove:

$$\lambda C(D_1, P_1) + (1 - \lambda)C(D_2, P_2) \geq C(\lambda D_1 + (1 - \lambda)D_2, \lambda P_1 + (1 - \lambda)P_2) \quad (8)$$

for any $\lambda \in [0, 1]$. First, let $\mu(\hat{x}|y)$ (resp. $\nu(\hat{x}|y)$) denote the optimal restoration method under constraint $(D_1, P_1)$ (resp. $(D_2, P_2)$), and $\hat{X}_\mu$ (resp. $\hat{X}_\nu$) be the restored signal, i.e.

$$\varepsilon(\hat{X}_\mu|c_0) = \min_{P_{\hat{X}|Y}} \varepsilon(\hat{X}|c_0), \text{subject to } \mathbb{E}[\Delta(X, \hat{X})] \leq D_1, d(p_X, p_{\hat{X}}) \leq P_1 \quad (9)$$

$$\varepsilon(\hat{X}_{\nu}|c_0) = \min_{P_{\hat{X}|Y}} \varepsilon(\hat{X}|c_0), \text{subject to } \mathbb{E}[\Delta(X,\hat{X})] \leq D_2, d(p_X, p_{\hat{X}}) \leq P_2 \qquad (10)$$

Then the left hand side of (8) becomes

$$\lambda\varepsilon(\hat{X}_{\mu}|c_0) + (1-\lambda)\varepsilon(\hat{X}_{\nu}|c_0) = \varepsilon(\hat{X}_{\lambda}|c_0) \qquad (11)$$

where $\hat{X}_{\lambda}$ denotes the restored signal corresponding to $p_{\lambda}(\hat{x}|y) = \lambda\mu(\hat{x}|y) + (1-\lambda)\nu(\hat{x}|y)$ (*see the supplementary for the proof of this equation*). Let $D_{\lambda} = \mathbb{E}[\Delta(X,\hat{X}_{\lambda})]$, $P_{\lambda} = d(p_X, p_{\hat{X}_{\lambda}})$, then by definition

$$\varepsilon(\hat{X}_{\lambda}|c_0) \geq \min_{P_{\hat{X}|Y}} \left\{ \varepsilon(\hat{X}|c_0) : \mathbb{E}[\Delta(X,\hat{X})] \leq D_{\lambda}, d(p_X, p_{\hat{X}}) \leq P_{\lambda} \right\} = C(D_{\lambda}, P_{\lambda}) \qquad (12)$$

Next, as $d(\cdot,\cdot)$ in (7) is convex in its second argument, we have

$$\begin{aligned} P_{\lambda} &= d(p_X, \lambda p_{\hat{X}_{\mu}} + (1-\lambda)p_{\hat{X}_{\nu}}) \\ &\leq \lambda d(p_X, p_{\hat{X}_{\mu}}) + (1-\lambda)d(p_X, p_{\hat{X}_{\nu}}) \\ &\leq \lambda P_1 + (1-\lambda)P_2 \end{aligned} \qquad (13)$$

the last inequality is due to (9) and (10). Similarly, we have

$$\begin{aligned} D_{\lambda} &= \mathbb{E}[\Delta(X,\hat{X}_{\lambda})] = \mathbb{E}_Y \mathbb{E}[\Delta(X,\hat{X}_{\lambda})|Y] \\ &= \mathbb{E}_Y[\lambda\mathbb{E}[\Delta(X,\hat{X}_{\mu})|Y] + (1-\lambda)\mathbb{E}[\Delta(X,\hat{X}_{\nu})|Y]] \\ &= \lambda\mathbb{E}[\Delta(X,\hat{X}_{\mu})] + (1-\lambda)\mathbb{E}[\Delta(X,\hat{X}_{\nu})] \\ &\leq \lambda D_1 + (1-\lambda)D_2 \end{aligned} \qquad (14)$$

the last inequality is again due to (9) and (10). Finally, note that $C(D,P)$ is non-increasing with respect to $D$ and $P$,

$$C(D_{\lambda}, P_{\lambda}) \geq C(\lambda D_1 + (1-\lambda)D_2, \lambda P_1 + (1-\lambda)P_2) \qquad (15)$$

Combining (11), (12), and (15), we have (8). $\qquad\square$

**Discussion.** Note that the property of the CDP function is quite similar to that of the perception-distortion function [3], and the proof is similar, too. The theorem has assumed the convexity of the function $d(\cdot,\cdot)$, which is satisfied by a large number of commonly used functions, including any f-divergence (e.g. KL, TV, Hellinger) and the Rényi divergence [4, 18]. The theorem does not require any assumption on the function $\Delta(\cdot,\cdot)$, implying that the CDP tradeoff exists for any distortion metric, including MSE/PSNR, SSIM, and the so-called feature losses which are calculated between deep features [8], and so on. The convexity of $C(D,P)$ implies the tradeoff is stronger at the low distortion or low perception regimes. In these regimes, any small improvement in distortion/perception achieved by a restoration algorithm, must be accompanied by a large loss of classification accuracy. Similarly, any small improvement in classification accuracy achieved by an algorithm whose error rate is already small, must be accompanied by a large increase of distortion and/or perceptual difference.

## 4 Experiments

In this section, we want to demonstrate the CDP tradeoff by real-world datasets and realistic settings. We use the MNIST handwritten digit recognition dataset [11] and the CIFAR-10 image recognition dataset [9]. The restoration tasks we considered are denoising and super-resolution (SR), and we use trained networks to perform the tasks. Since our intention is not to study the restoration method itself, we design simple denoising and SR networks inspired by the successful DnCNN [23] and SRCNN [6],

Table 1: Experimental configurations. CNN-2 and CNN-2' have the same network structure but differ in input image size ($28\times28$ and $32\times32$).

|       | Dataset  | Task      | Classifier |
|-------|----------|-----------|------------|
| Exp-1 | MNIST    | Denoising | Logistic   |
| Exp-2 | MNIST    | Denoising | CNN-1      |
| Exp-3 | MNIST    | Denoising | CNN-2      |
| Exp-4 | MNIST    | SR        | CNN-1      |
| Exp-5 | CIFAR-10 | SR        | CNN-2'     |

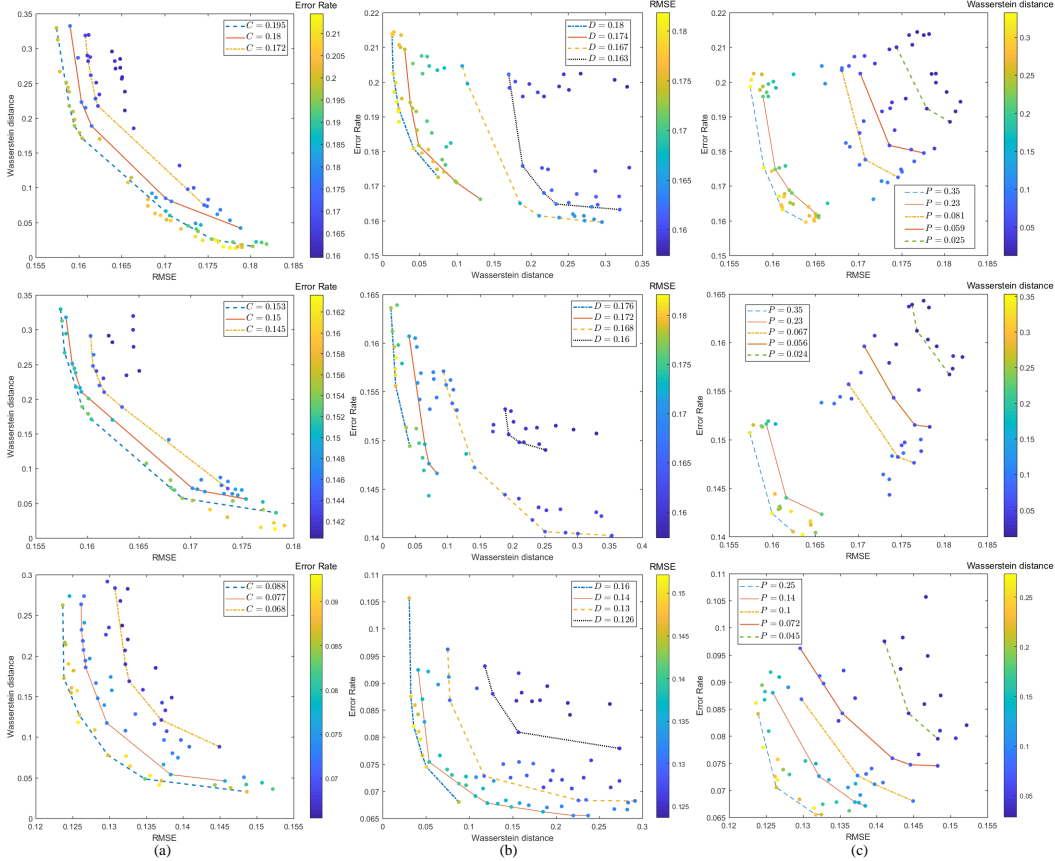

Figure 3: Profiles of the CDP functions. From top to bottom: Exp-1, Exp-2, and Exp-4. Better classification performance always comes at the cost of higher distortion and worse perceptual quality.

respectively. Experimental configurations are summarized in Table 1. More details can be found in the supplementary.

In order to showcase the CDP tradeoff, we train a restoration (denoising or SR) network with a combination of three loss functions that correspond to distortion, perceptual difference, and classification error rate. In short, the entire loss function is

$$\ell_{restoration} = \alpha \ell_{MSE} + \beta \ell_{adv} + \gamma \ell_{CE} \tag{16}$$

where $\alpha, \beta, \gamma$ are weights. The first term is MSE loss to represent distortion, which is widely used in image restoration research. The second term is an adversarial loss, minimizing which is to ensure perceptual quality as suggested in [3]. Here we adopt the Wasserstein GAN [1] and the adversarial loss $\ell_{adv}$ is proportional to the Wasserstein distance $d_W(p_X, p_{\hat{X}})$. Note that in the Wasserstein GAN, the discriminator loss is indeed an estimate of the Wasserstein distance, which can be used to assess the perceptual quality of the restored images quantitatively. The third term is cross entropy, corresponding to classification error rate. To demonstrate that the CDP tradeoff is generic, we use multiple classifiers in experiments: the first is a simple logistic regression, and the others are CNN-based classifiers. For each classifier, we pretrain it on the clean (i.e. noise-free and original-resolution) training data, and use it to evaluate cross entropy when training the denoising or SR network.

For Exp-1, Exp-2, and Exp-3, noisy images are generated by adding Gaussian noise $\mathcal{N}(0,1)$ onto the MNIST images. Then, the noisy training data as well as their clean version are used to train the denoiser, with different combinations of $(\alpha, \beta, \gamma)$. After training we use the denoiser to process the noisy MNIST test data, and calculate D (MSE), P (Wasserstein distance using the discriminator), and C (using the pretrained classifier). For Exp-4, MNIST images are down-sampled by a factor of 6 and then interpolated to original resolution. Interpolated images and their clean version are used to train the SR network. For Exp-5, CIFAR-10 images are down-sampled by a factor of 3.

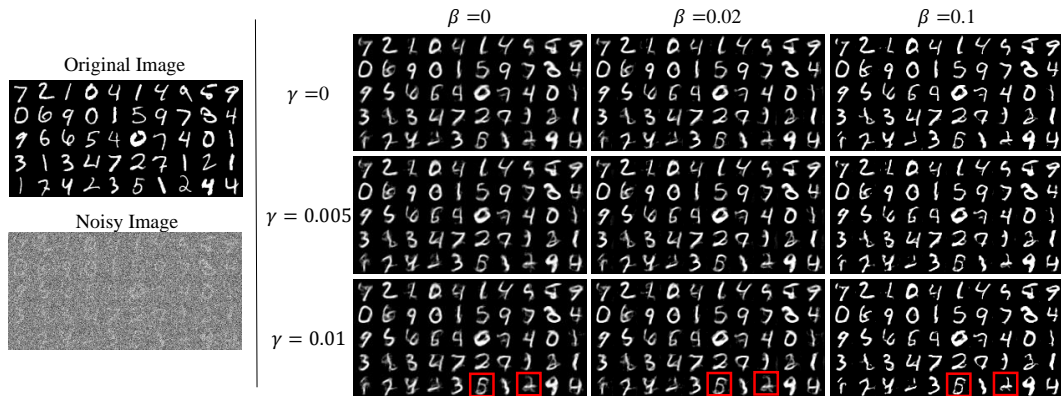

Figure 4: Visual results of Exp-2 with different combinations of loss weights. As $\gamma$ increases, the perceptual quality becomes worse but the restored images are easier to recognize, see for example the numbers '5' and '2' highlighted by red boxes.

Fig. 3 presents the results of Exp-1/2/4, where we plot each pair of (C, D, P) separately. For example in (a), we plot the relation of P and D and using color to denote C. We also draw curves to connect the points with approximately the same C. As can be observed, when C is sufficiently large, there is a tradeoff between P and D, which has been characterized in [3]. Once C is smaller, the P-D curve elevates, indicating that better classification performance comes at the cost of higher distortion and/or worse perceptual quality. Similarly in (b) and (c), we can observe the relations of C-P and C-D and all of them are convex as the theorem forecasts. Moreover, comparing Exp-1 and Exp-2 that use different classifiers, although the error rates differ much in number, the trends of the CDP tradeoff are similar. Please check the supplementary for more results.

Fig. 4 presents some results of Exp-2 for visual inspection. As observed, the visual quality of denoised images in general increases along with the weight $\beta$. Given the same $\beta$, when increasing $\gamma$, the visual quality decreases, showing a tradeoff. As expected, increasing $\gamma$ will enhance the semantic quality of the denoised images, which is actually evaluated by the pretrained classifier. Please note the numbers '5' and '2' highlighted by red boxes, these numbers may be difficult to recognize if $\gamma$ is small, but seem recognizable when $\gamma$ is large. There seems a positive correlation between classification error rate (which is evaluated by the classifier) and human recognition (which is evaluated by ourselves). Note that the human recognition is different from the visual quality: human recognition means whether the class can be correctly recognized by human, visual quality (perceptual naturalness as defined in this paper) means whether the image looks like a natural image. More visual examples are provided in the supplementary.

# 5 Conclusion

We have addressed the classification-distortion-perception tradeoff by both proving a theorem about the characteristic of the CDP function and showcasing the CDP functions under simulation and experimental settings. Regardless of the restoration algorithm, the classification error rate on the restored signal evaluated by a predefined classifier cannot be made minimal along with the distortion and perceptual difference. The CDP function is convex, indicating that when the error rate is already low, any improvement of classification performance comes at the cost of higher distortion and worse perceptual quality.

Our findings can be useful especially for computer vision research where some low-level vision tasks (signal restoration) serve for high-level vision tasks (visual understanding). It is worth noting that we have used a predefined classifier to evaluate the classification error rate, but in practice, we may have a different metric that directly measures the semantic quality of restored signal. More studies are expected at this aspect in the future.

**Acknowledgments**

This work was supported by the Natural Science Foundation of China under Grant 61772483.

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
