[Supplementary Material]

# Supplementary Material for
# On The Classification-Distortion-Perception Tradeoff

**Dong Liu, Haochen Zhang, Zhiwei Xiong**
University of Science and Technology of China, Hefei 230027, China
dongeliu@ustc.edu.cn

This supplementary material consists of four parts. First, we prove two theorems about the properties of classification error rate (used in Section 2.2 and for Eq. (11)). Second, we present the calculation of distortion and classification error rate for the toy example (used in Section 2.3). Third, we describe the experimental details (used in Section 4). Fourth, we report more experimental results.

## A Properties of the Classification Error Rate

### A.1 Bayes Error Rate is Non-Decreasing

In Section 2.2 we mentioned that, if we have known the distribution of the degraded signal $Y$, then signal restoration has no help for classification, because the optimal classifier for $\hat{X}$ cannot outperform the optimal classifier for $Y$. Here we provide the proof.

To begin, it is necessary to define what is the optimal classifier. The optimal classifier is defined as the classifier that achieves the minimal error rate for a given signal, e.g. $c_X^* = \arg\min_c \varepsilon(X|c)$. According to the Bayes decision rule (see [Fuk90] for proof), the optimal classifier shall be

$$c_X^* = c(\cdot|\mathcal{R}_X^*), \text{ where } \mathcal{R}_X^* = \{x|P_1 p_{X1}(x) \geq P_2 p_{X2}(x)\} \tag{a}$$

which leads to the minimal error rate, a.k.a. the Bayes error rate

$$
\begin{aligned}
\epsilon(X) = \min_c \varepsilon(X|c) &= \varepsilon(X|\mathcal{R}_X^*) \\
&= \sum_x \min[P_1 p_{X1}(x), P_2 p_{X2}(x)] \\
&= \frac{1}{2} - \frac{1}{2}\sum_x |P_1 p_{X1}(x) - P_2 p_{X2}(x)|
\end{aligned}
\tag{b}
$$

Then we are able to prove a quite general theorem, which claims that the Bayes error rate will not decrease after any data processing step (surely including signal restoration).

**Theorem a.** *Let the process of $X \to Y$ be denoted by $P_{Y|X}$, which is characterized by a conditional mass function $p(y|x)$, then $\epsilon_Y \geq \epsilon_X$. $\epsilon_Y = \epsilon_X$ if and only if $p(y|x)$ satisfies: $\forall x_1 \in \mathcal{R}^+, \forall x_2 \in \mathcal{R}^-, \forall y, p(y|x_1)p(y|x_2) = 0$, where $\mathcal{R}^+ = \{x|P_1 p_{X1}(x) > P_2 p_{X2}(x)\}$, and $\mathcal{R}^- = \{x|P_1 p_{X1}(x) < P_2 p_{X2}(x)\}$.*

*Proof.*

$$\epsilon_Y = \sum_y \min[P_1 p_{Y1}(y), P_2 p_{Y2}(y)]$$

$$= \frac{1}{2} - \frac{1}{2}\sum_y |P_1 p_{Y1}(y) - P_2 p_{Y2}(y)|$$

$$= \frac{1}{2} - \frac{1}{2}\sum_y \left| P_1 \sum_x p(y|x)p_{X1}(x) - P_2 \sum_x p(y|x)p_{X2}(x) \right|$$

$$= \frac{1}{2} - \frac{1}{2}\sum_y \left| \sum_x p(y|x)[P_1 p_{X1}(x) - P_2 p_{X2}(x)] \right| \qquad \text{(c)}$$

$$\geq \frac{1}{2} - \frac{1}{2}\sum_y \sum_x p(y|x)|P_1 p_{X1}(x) - P_2 p_{X2}(x)|$$

$$= \frac{1}{2} - \frac{1}{2}\sum_x |P_1 p_{X1}(x) - P_2 p_{X2}(x)| \sum_y p(y|x)$$

$$= \frac{1}{2} - \frac{1}{2}\sum_x |P_1 p_{X1}(x) - P_2 p_{X2}(x)| = \epsilon_X$$

When $\epsilon_Y = \epsilon_X$, for any $y$, we need to have

$$\left| \sum_x p(y|x)[P_1 p_{X1}(x) - P_2 p_{X2}(x)] \right| = \sum_x p(y|x)|P_1 p_{X1}(x) - P_2 p_{X2}(x)| \qquad \text{(d)}$$

which is equivalent to: all the $x$'s that satisfy $p(y|x) \neq 0$ shall have either $P_1 p_{X1}(x) - P_2 p_{X2}(x) \geq 0$ or $P_1 p_{X1}(x) - P_2 p_{X2}(x) \leq 0$. The condition is further equivalent to: the $x$'s that satisfy $p(y|x) \neq 0$ shall be either all in $\mathcal{R}^+ \cup \mathcal{R}^0$, or all in $\mathcal{R}^- \cup \mathcal{R}^0$, where $\mathcal{R}^0 = \{x | P_1 p_{X1}(x) = P_2 p_{X2}(x)\}$. In other words, $\forall x_1 \in \mathcal{R}^+, \forall x_2 \in \mathcal{R}^-, p(y|x_1)p(y|x_2) = 0$. □

We can compare Theorem a with the data processing theorem in the information theory: consider the process of $X \rightarrow Y$ as a deterministic function $Y = f(X)$, then $I(X;Y) \leq H(X)$, and $I(X;Y) = H(X)$ if and only if $f$ is invertible [CT12]. That says, the information quantity we have about the source $X$ is non-increasing after data processing. Similarly, Theorem a claims that the Bayes error rate is non-decreasing after data processing, *because we lose information, at best none*. Moreover, not only invertible function satisfies the condition required in Theorem a, but also a large group of non-invertible functions as well as probabilistic mappings satisfy the condition, which is quite different from the data processing theorem. In other words, *we may lose information but that information loss may not affect classification*.

Given Theorem a, we conclude that if we are dealing with non-blind restoration, i.e. the distribution of the degraded signal is known, then we can achieve the optimal classifier for the degraded signal, and it is not necessary to perform signal restoration prior to classification as it will not improve the classification performance. However, blind restoration is a more appealing setting in practice since the degradation process is often unknown. In this paper, we are interested more in the blind restoration case.

## A.2 Classification Error Rate is Linear

During the proof of Theorem 1, we have used the linear property of the classification error rate given a fixed classifier. This property is ensured by the following theorem.

**Theorem b.** *Let $U$ follow a two-component mixture model: $p_U(u) = P_1 p_{U1}(u) + P_2 p_{U2}(u)$, similarly $V$ follow: $p_V(v) = P_1 p_{V1}(v) + P_2 p_{V2}(v)$. Let $W$ be the random variable with $p_W(w) = \lambda p_U(w) + (1-\lambda)p_V(w)$ where $0 \leq \lambda \leq 1$. Let $c_0$ be a fixed classifier, then*

$$\varepsilon(W|c_0) = \lambda \varepsilon(U|c_0) + (1-\lambda)\varepsilon(V|c_0) \qquad \text{(e)}$$

*Proof.* As $c_0$ is a fixed classifier, it can be denoted in general by $c_0 = c(\cdot|\mathcal{R}_0)$. Then we have

$$\varepsilon(U|c_0) = P_2 \sum_{u \in \mathcal{R}_0} p_{U2}(u) + P_1 \sum_{u \notin \mathcal{R}_0} p_{U1}(u) \tag{f}$$

$$\varepsilon(V|c_0) = P_2 \sum_{v \in \mathcal{R}_0} p_{V2}(v) + P_1 \sum_{v \notin \mathcal{R}_0} p_{V1}(v) \tag{g}$$

Thus

$$
\begin{aligned}
\varepsilon(W|c_0) &= P_2 \sum_{w \in \mathcal{R}_0} p_{W2}(w) + P_1 \sum_{w \notin \mathcal{R}_0} p_{W1}(w) \\
&= P_2 \sum_{w \in \mathcal{R}_0} [\lambda p_{U2}(w) + (1-\lambda)p_{V2}(w)] + P_1 \sum_{w \notin \mathcal{R}_0} [\lambda p_{U1}(w) + (1-\lambda)p_{V1}(w)] \\
&= \lambda \left[ P_2 \sum_{w \in \mathcal{R}_0} p_{U2}(w) + P_1 \sum_{w \notin \mathcal{R}_0} p_{U1}(w) \right] + (1-\lambda) \left[ P_2 \sum_{w \in \mathcal{R}_0} p_{V2}(w) + P_1 \sum_{w \notin \mathcal{R}_0} p_{V1}(w) \right] \\
&= \lambda \varepsilon(U|c_0) + (1-\lambda)\varepsilon(V|c_0)
\end{aligned}
\tag{h}
$$

$\square$

## B  Calculation for the Toy Example

Let us review the settings of the toy example: $p_X(x) = P_1 p_{X1}(x) + P_2 p_{X2}(x)$, where $p_{X1}(x) = \mathcal{N}(-1, 1)$, $p_{X2}(x) = \mathcal{N}(1, 1)$. $Y = X + N$ where $N \sim \mathcal{N}(0, \sigma_N^2)$. $\hat{X} = aY$ where $a$ is an adjustable parameter. we have $p_{\hat{X}}(x) = P_1 p_{\hat{X}1}(x) + P_2 p_{\hat{X}2}(x)$, where $p_{\hat{X}1}(x) = \mathcal{N}[-a, a^2(1 + \sigma_N^2)]$, $p_{\hat{X}2}(x) = \mathcal{N}[a, a^2(1 + \sigma_N^2)]$.

Since $N$ is independent from $X$ and $\mathbb{E}(X^2) = P_1 \mathbb{E}(X_1^2) + P_2 \mathbb{E}(X_2^2)$,

$$\text{MSE}(a) = \mathbb{E}[(X - \hat{X})^2] = \mathbb{E}(X^2) + \mathbb{E}(\hat{X}^2) - 2\mathbb{E}(X\hat{X}) = (2 + \sigma_N^2)a^2 - 4a + 2 \tag{i}$$

According to the Bayes decision rule, the optimal classification plane for the original signal can be obtained by solving $P_1 p_{X1}(x_0) = P_2 p_{X2}(x_0)$, which leads to $x_0 = -\frac{1}{2}\ln\frac{P_2}{P_1}$. Applying this classifier on the denoised signal $\hat{X}$, the error rate will be

$$
\begin{aligned}
\varepsilon(\hat{X}|c_0) &= P_2 \int_{-\infty}^{x_0} \mathcal{N}[a, a^2(1 + \sigma_N^2)]dx + P_1 \int_{x_0}^{\infty} \mathcal{N}[-a, a^2(1 + \sigma_N^2)]dx \\
&= P_2 \int_{-\infty}^{x_0'} \mathcal{N}(0, 1)dx + P_1 \int_{x_0''}^{\infty} \mathcal{N}(0, 1)dx \\
&= P_2 \Phi(x_0') + P_1 \Phi(-x_0'')
\end{aligned}
\tag{j}
$$

where $x_0' = \frac{x_0 - a}{|a|\sqrt{1+\sigma_N^2}}$ and $x_0'' = \frac{x_0 + a}{|a|\sqrt{1+\sigma_N^2}}$. $\Phi(\cdot)$ is the integral of standard normal distribution.

For the perception we cannot derive a closed-form function, so we use the numerical method to calculate.

Note that distortion, perception, and error rate are all determined by $a$. According to the CDP function, if given a constraint on $D$ (or $P$), the feasible domain of $a$ is restricted, and then the minimal reachable value of the error rate is also restricted, which is essentially the tradeoff.

## C  Experimental Details

We use TensorFlow for implementation. The network structures used in our experiments are listed in Table a.

**Exp-1.** First, we use the clean (i.e. noise-free) MNIST training data to train a logistic regression classifier. When tested on the clean MNIST test set, the logisitc regression classifier achieves 92.73%

Table a: Network structures. Conv(kernel size, stride, channel, padding) stands for a convolutional layer with the corresponding settings (padding s stands for same and v stands for valid). BN is batch normalization, L-ReLU is Leaky-ReLU with slope 0.2, MP is max pooling where stride is 2, Dropout is with probability 0.5. For MNIST, H=W=28, C=1; for CIFAR-10, H=W=32, C=3.

| Denoiser (Exp-1/2/3) |
|---|
| Input (H×W×C) |
| Conv(5, 1, 32, s)+ReLU |
| Conv(5, 1, 64, s)+BN+ReLU |
| Conv(5, 1, 64, s)+BN+ReLU |
| Conv(5, 1, 32, s)+BN+ReLU |
| Conv(5, 1, C, s)+clip to [0,1] |
| Output (H×W×C) |

| SR Network (Exp-4/5) |
|---|
| Input (H×W×C) |
| Conv(9, 1, 64, s)+ReLU |
| Conv(5, 1, 32, s)+ReLU |
| Conv(5, 1, C, s)+clip to [0,1] |
| Output (H×W×C) |

| Discriminator (All Exps) |
|---|
| Input (H×W×C) |
| Conv(5, 2, 32, s)+L-ReLU |
| Conv(5, 2, 64, s)+BN+L-ReLU |
| Conv(5, 2, 128, s)+BN+L-ReLU |
| FC (output=1) |
| Output (1) |

| CNN-1 Classifier (Exp-2/4) |
|---|
| Input (H×W×C) |
| Conv(5, 1, 10, v)+MP+ReLU |
| Conv(5, 1, 20, v)+MP+ReLU |
| FC (output=50)+Dropout |
| FC (output=10) |
| Output (10) |

| CNN-2(2') Classifier (Exp-3/5) |
|---|
| Input (H×W×C) |
| Conv(3, 1, 64, s)+ReLU |
| Conv(3, 1, 64, s)+ReLU+MP |
| Conv(3, 1, 64, s)+ReLU |
| Conv(3, 1, 64, s)+ReLU+MP |
| Conv(3, 1, 64, s)+ReLU |
| Conv(3, 1, 64, s)+ReLU+MP |
| FC (output=256)+ReLU+Dropout |
| FC (output=10) |
| Output (10) |

Table b: Loss weight combinations for Exp-1.

| $\gamma$ | $\alpha$ | $\beta$ |
|---|---|---|
| 0 | 0.9 | 0.0175, 0.0195 |
| | 0.95 | 0.0178 |
| | 1 | 0, 0.004, 0.008, 0.0165, 0.017, 0.0172, 0.018, 0.019, 0.024, 0.032, 0.04, 0.06, 0.08, 0.1, 0.2, 0.3 |
| | 1.1 | 0.0175 |
| 0.005 | 1 | 0, 0.016, 0.0188, 0.019, 0.0197, 0.0198, 0.0199, 0.0202, 0.0212, 0.028, 0.032, 0.034, 0.05, 0.2, 0.3 |
| 0.01 | 1 | 0.0118, 0.0122, 0.013, 0.0138, 0.0142, 0.015, 0.02, 0.0203, 0.0205, 0.021, 0.0212, 0.0213, 0.0264, 0.032, 0.1, 0.2, 0.3 |
| 0.02 | 1 | 0.015, 0.02, 0.022, 0.024, 0.0244, 0.0248, 0.025, 0.0251, 0.0252, 0.0253, 0.0254, 0.0255, 0.027, 0.0275, 0.028, 0.0295, 0.03, 0.032, 0.038, 0.05, 0.1, 0.2, 0.3 |

| $\gamma$ | $\beta$ | $\alpha$ |
|---|---|---|
| 0 | 0.0165 | 0.92, 0.931, 0.935, 0.937 |
| | 0.017 | 0.96, 0.961, 0.962, 0.97, 1.1 |

Table c: Loss weight combinations for Exp-2.

| $\gamma$ | $\alpha$ | $\beta$ |
|---|---|---|
| 0 | 0.9 | 0.0165, 0.0171, 0.0175, 0.0185, 0.019, 0.0195 |
| | 0.95 | 0.0167, 0.017, 0.0172, 0.0173 |
| | 1 | 0, 0.004, 0.008, 0.0165, 0.017, 0.0172, 0.0174, 0.0178, 0.018, 0.0185, 0.019, 0.0195, 0.02, 0.024, 0.032, 0.04, 0.06, 0.08, 0.1, 0.2, 0.3 |
| | 1.1 | 0.017, 0.0175, 0.0185, 0.019, 0.0195 |
| 0.000625 | 1 | 0, 0.000375 |
| 0.005 | 1 | 0, 0.016, 0.018, 0.0206, 0.0208, 0.021, 0.0214, 0.0216, 0.022, 0.024, 0.026, 0.028, 0.1, 0.3 |
| 0.01 | 0.95 | 0.0252, 0.0254 |
| | 1 | 0, 0.01, 0.015, 0.02, 0.022, 0.024, 0.0255, 0.0265, 0.027, 0.0275, 0.028, 0.03, 0.032, 0.034, 0.036, 0.038, 0.05, 0.1, 0.2, 0.3 |
| | 1.02 | 0.0261, 0.0262, 0.0263, 0.0264 |
| 0.015 | 1 | 0, 0.0252, 0.0258, 0.0276, 0.028, 0.0282, 0.03, 0.032, 0.1, 0.3 |

| $\gamma$ | $\beta$ | $\alpha$ |
|---|---|---|
| 0 | 0.0165 | 0.93, 0.931, 0.935, 0.937, 0.94 |
| | 0.017 | 0.96, 0.961, 0.962, 0.965, 0.969, 0.97, 0.99 |
| 0.005 | 0.0212 | 0.92, 0.94, 0.96 |
| | 0.0214 | 1.02, 1.04, 1.06 |
| 0.01 | 0.026 | 0.97, 1.018 |
| | 0.0262 | 1.03, 1.07 |
| 0.015 | 0.0294 | 0.92, 0.94, 0.98 |

Table d: Loss weight combinations for Exp-3.

| $\alpha$ | $\gamma$ | $\beta$ |
|---|---|---|
| 1 | 0 | 0, 0.011, 0.012, 0.014, 0.015, 0.016, 0.01784, 0.01786, 0.01788, 0.0179, 0.01792, 0.01794, 0.01796, 0.018, 0.019, 0.02, 0.03, 0.0362, 0.0366, 0.0368, 0.06, 0.08, 0.1 |
| | 0.001 | 0, 0.011, 0.012, 0.013, 0.015, 0.016, 0.017, 0.018, 0.0186, 0.0188, 0.01882, 0.01886, 0.0189, 0.01894, 0.019, 0.02, 0.03, 0.04 |
| | 0.003 | 0, 0.021, 0.022, 0.023, 0.02302, 0.02306, 0.02308, 0.02314, 0.02316, 0.0232, 0.0234, 0.024, 0.027, 0.028, 0.03, 0.04 |
| | 0.005 | 0.01, 0.02, 0.021, 0.024, 0.025, 0.026, 0.028, 0.029, 0.03, 0.04 |
| | 0.008 | 0.0312, 0.0314, 0.033, 0.035 |

Table e: Loss weight combinations for Exp-4.

| $\alpha$ | $\gamma$ | $\beta$ |
|---|---|---|
| 1 | 0 | 0, 0.001, 0.002, 0.003, 0.004, 0.006, 0.009, 0.01, 0.011, 0.012, 0.013, 0.015, 0.016, 0.018, 0.019, 0.02, 0.03, 0.04, 0.1, 0.15, 0.2 |
| | 0.001 | 0, 0.01, 0.03, 0.04, 0.15 |
| | 0.0025 | 0, 0.02 , 0.1 |
| | 0.005 | 0, 0.001, 0.002, 0.003, 0.004, 0.006, 0.009, 0.01, 0.012, 0.013, 0.014, 0.015, 0.016, 0.018, 0.03, 0.04, 0.1, 0.15, 0.2 |
| | 0.01 | 0, 0.001, 0.002, 0.02, 0.024, 0.026, 0.03, 0.04, 0.1, 0.15, 0.2 |
| | 0.015 | 0, 0.001, 0.004, 0.006, 0.011, 0.014, 0.016, 0.02, 0.0225, 0.023, 0.024, 0.025, 0.027, 0.03, 0.04, 0.1, 0.15, 0.2 |
| | 0.02 | 0.0225, 0.0235, 0.024, 0.025, 0.03 |

Table f: Loss weight combinations for Exp-5.

| $\alpha$ | $\gamma$ | $\beta$ |
|---|---|---|
| 1 | 0 | 0.008, 0.009, 0.011, 0.012, 0.014, 0.015, 0.017, 0.019, 0.02, 0.03, 0.05, 0.06, 0.1 |
| | 0.00001 | 0.008, 0.009, 0.01, 0.011, 0.012, 0.013, 0.015, 0.016, 0.017, 0.018, 0.019, 0.02, 0.03, 0.04, 0.05, 0.07, 0.08, 0.1 |
| | 0.0002 | 0.008, 0.009, 0.011, 0.014, 0.016 |
| | 0.0003 | 0.007, 0.009, 0.011, 0.013, 0.017, 0.019 |
| | 0.0005 | 0.006, 0.007, 0.008, 0.009, 0.01, 0.011, 0.012, 0.013, 0.016, 0.017, 0.019, 0.03, 0.04, 0.06, 0.07, 0.08, 0.1 |
| | 0.0006 | 0.007, 0.008, 0.009, 0.01, 0.011, 0.013, 0.016, 0.017, 0.019, 0.03, 0.04, 0.05 |

accuracy, which we think is satisfactory on MNIST. Then, we fix the classifier and train the denoiser and discriminator in an adversarial manner as suggested in [ACB17]. Note that the training images are already corrupted by noise $\mathcal{N}(0, 1)$. For denoiser, the loss is $\ell_{denoiser} = \alpha\ell_{MSE} + \beta\ell_{adv} + \gamma\ell_{class}$, where $\ell_{MSE}$ is the MSE loss, $\ell_{adv}$ is the adversarial loss [ACB17], and $\ell_{class}$ is the classification loss (i.e. cross entropy calculated by the pretrained classifier). $\alpha, \beta, \gamma$ are the weights and we have tried many different combinations of weights as listed in Table b. We use the ADAM optimizer with hyper-parameters $\beta_1 = 0.5, \beta_2 = 0.9$. The initial learning rate of denioser/discriminator is $10^{-3}/10^{-4}$, and the learning rate of denoiser decreases to $1/5$ every 10,000 iterations. Batch size is 50 and training stops at 40,000 iterations. Model selection is performed by using the MNIST validation set.

**Exp-2.** Different from Exp-1, we use a CNN-based classifier, namely CNN-1 in Table a. We use the clean MNIST training data and the SGD optimizer with a constant learning rate 0.01 to train CNN-1. Batch size is 100 and training stops at 140 epoch. As a result, we achieve 99.19% accuracy on the clean MNIST test set. Then, we fix the classifier and train the denoiser and discriminator. Loss function is the same, but the cross entropy is calculated by the CNN-1 classifier. Combinations of weights are listed in Table c.

**Exp-3**. Almost identical to Exp-2, but using another CNN-based classifier, namely CNN-2 in Table a. Learning rate is set as 0.1/0.01/0.002/0.001/0.0001 at the beginning of 1/101/201/501/1501 epoch and training stops at 2000 epoch. On the clean MNIST test set, CNN-2 performs slightly better than CNN-1, i.e. the accuracy is 99.38%. Then we fix the classifier and train the denoiser and discriminator. Combinations of weights are listed in Table d.

**Exp-4**. The classifier is CNN-1 which is already trained in Exp-2. We fix the classifier and train the SR network and discriminator. Combinations of weights are listed in Table e.

**Exp-5**. We use CNN-2', whose structure is almost identical to CNN-2 but input image size is different. We use the raw CIFAR-10 training data to train CNN-2' (with data augmentation such as horizontal flipping and padding + random cropping), achieving 85.18% accuracy on the raw CIFAR-10 test set. Then we fix the classifier and train the SR network and discriminator. Combinations of weights are listed in Table f.

Figure a: Profiles of the CDP functions. Top: Exp-3, middle: Exp-5, bottom: Exp-4 displayed in another style, where the size of each point indicates the corresponding value (quantized).

## D   More Experimental Results

Profiles of the CDP functions of Exp-3/4/5 are shown in Fig. a. Some visual examples of Exp-4 and Exp-5 are shown in Fig. b.

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

Figure b: Visual results. Top: Exp-4, bottom: Exp-5.