[Reviews · NeurIPS 2019]

Reviewer 1



Updated Review: Thanks to the authors for fixing the plots, the intended message of those plots is now much easier to ascertain. In light of the author response I have decided to keep my review of this work the same, as I believe the authors have successfully demonstrated that the work is of high quality. Originality: The work builds off of a recently published result, the perception-distortion tradeoff, and shows that the original work was incomplete as it did not take into account the third dimension of the tradeoff, classification accuracy. Though the idea of a tradeoff between objectives in signal restoration is not novel, the work is a novel contribution and extension to the theory around this important area of research. Quality: The work is overall of very high quality, with a combination of rigorous proofs, experimental verification, and sound justification for the approach. Clarity: I find the plots in figure 3 to be difficult to easily ascertain the fundamental idea from (though with some effort I understand what they are conveying). I recognize the authors attempt to recreate the plot style from the original perception-distortion paper. Perhaps the authors can find a more intelligible way to break down the many dimensions being displayed in those plots. The paper is otherwise clear and well argued. Significance: The paper strikes me as an extremely significant followup to a recent paper, and relevant to a wide audience at NeurIPS as signal restoration is a topic of high interest, especially in safety critical domains. Better knowledge of these fundamental limitations and trade-offs will help many engineers design for the correct objectives for their applications.

Reviewer 2



originality: the formal justification of the CDP trade-off is a novel contribution quality: the derivation of the CDP function is technically sound. While inspiring, the empirical justification could be improved: 1) results are only shown on mnist, which is a set of close-to-binary images. How does the CDP tradeoff look like on more general natural images? 2) only one simple degradation scenario of additive Gaussian noise is considered. How does the CDP tradeoff performs beyond this, in the presence other degradations, e.g. blur, low-res, exposure? clarify: the presentation of the paper is clear significance: the general trade-off between classification-distortion-perception is of broad interest to the community of computer vision and machine learning ** Updates ** The authors have partially addressed the concern on generalization beyond simple additive white Gaussian noise degradation, and promise to conduct analysis on datasets beyond mnist. Overall, the formulation and analysis of CDP is a good contribution to the community.

Reviewer 3



This work is a novel extension of the preceding distortion-perception tradeoff in image restoration tasks. The practicality is enhanced by incorporating analysis of classification errors in related tasks. The claims of this work are technically sound and supported by comprehensive analysis and the experiments. The proof is straightforward and clear. Analysis on the toy example gives solid evidence to support the claim. The overall structure of this paper is clear and easy to follow. Most parts of this work are clearly explained. Issues: - There seems to be an ambiguity between the proposed CDP tradeoff and the discussion in section 4. Specifically, the described CDP suggests that classification error and perceptual naturalness cannot be simultaneously optimized. Yet, in section 4 (line 247-251), the authors also suggested a positive correlation between the classification error rate and "subjective" feeling. This is confusing as it implies the opposite to the original argument. The authors might want to provide a clarification about the statement in section 4 to avoid confusion. - Visual analysis of the experiments can be more thorough. The authors only provided visual results to compare the classification-perception tradeoff with some analysis. While this is not a major concern, it would be more informative if the authors can also provided similar comparisons for distortion and classification. The analysis on visual results can also be more carefully formulated, such that the results can pose a stronger argument. ---------------- I have read the author rebuttal and other reviewers' comments and I tend to recommend this paper. It will be great if the authors can provide more analysis and experimental results beyond MNIST.

[Author Response · NeurIPS 2019]

# Response Letter for Paper ID 733

## *On The Classification-Distortion-Perception Tradeoff*

We are grateful to the three reviewers for the time and effort in reviewing this paper, and for the recognition of the originality, quality, clarity, and significance of this paper. We will improve the paper according to the comments.

**Reviewer 1**

**Plots in Figure 3**. Thank you for the suggestion. We try a new plot style, as shown in Figure I (bottom left). We will provide new plots in this style in the revised paper or in the supplementary.

**Using different classifiers**. In Figure 3, we present the results of two classifiers: a CNN-based classifier and a logistic regression classifier. Different classifiers indeed lead to different tradeoff boundaries (please note the values of error rate). To address your comment, we plan to use a third classifier and redo the experiment. Due to limited time of rebuttal, that experiment cannot be finished now. We will add the experimental results into the revised paper.

**Reviewer 2**

**Results on general natural images**. We agree with you that more experimental results can make the conclusion more solid. We plan to test on the CIFAR-10 dataset. Due to limited time of rebuttal, that experiment cannot be finished now. We will add the experimental results into the revised paper.

**Results under other degradations**. Thank you for the suggestion. We conduct a new experiment about image super-resolution (SR) on the MNIST dataset. Original images are down-sampled by bicubic with a factor of 6. We use the structure of the well-known SRCNN [Dong *et al.*, ECCV 2014] for training SR networks. Other settings are the same as for the denoising experiment (using the same pretrained CNN-based classifier). The results are shown in Figure I (top row). They can confirm the conclusion drawn in the paper.

**Reviewer 3**

**Clarification about the statement in Section 4 (line 247-251)**. Thank you for the suggestion. In line 247-251 we discuss the correlation between classification error rate (which is evaluated by a classifier) and human recognition (which is evaluated by ourselves). The human recognition is different from the visual quality: human recognition means whether the class can be correctly recognized by human, visual quality (perceptual naturalness as defined in this paper) means whether the image looks like a natural image. We will revise the wording to avoid confusion.

**Visual comparisons for distortion and classification**. We will include more visual results in the revised paper or in the supplementary, such as CIFAR-10 results and SR results. As an example, please check Figure I (bottom right).

Figure I: Top: profiles of the CDP function for MNIST and SR. Bottom left: using a new style for the top left plot, where the size of each point indicates the corresponding error rate (quantized). Bottom right: some visual results to display the C-D tradeoff, where red boxes indicate examples that appear more recognizable by human.

[Meta-Review · NeurIPS 2019]

This submission received diverging scores and after rebuttal and discussion phase the review scores are 9,6,6. All reviewers recommend acceptance of the paper. There is consensus between the reviewers that this paper represents a contribution to NeurIPS. The submission presents an extension of the recent distortion-perception (DP) trade-off, including classification (CDP). This function presents a contribution to the task of image restoration and can be used for different tasks. The main contribution is the extension from DP to CDP, the proof is well presented and a toy example presents insights and understanding. The main points for improvements as pointed out by the reviewers are experiments over MNIST and denoising. Both points are answered in the rebuttal, please include those in a final revision. - Promised experiments for CIFAR-10. - Additional results beyond image denoising. The rebuttal presents initial results for super-resolution in line with claimed statements in the paper. All reviewers recommend acceptance and with the promised additions from the rebuttal this is a good contribution that is of interest to NeurIPS and beyond. Please make sure to include all other reviewer comments as well in the final version.